# The Mont Blanc Study: The effect of altitude on intra ocular pressure and central corneal thickness

Carlo Bruttini[1], Alice Verticchio Vercellin[1,2,3], Catherine Klersy[4], Annalisa De Silvestri[4], Carmine Tinelli[4], Ivano Riva[2], Francesco Oddone[2], Andreas Katsanos[5], Luciano Quaranta[1]*

1 University Eye Clinic, Fondazione IRCCS Policlinico San Matteo, Pavia, Italy, 2 IRCCS—Fondazione G.B. Bietti, Rome, Italy, 3 Department of Ophthalmology, Icahn School of Medicine at Mount Sinai Hospital, New York, NY, United States of America, 4 Clinical Epidemiology and Biometry Unit, Fondazione IRCCS Policlinico San Matteo, Pavia, Italy, 5 Department of Ophthalmology, University of Ioannina, Ioannina, Greece

* luciano.quaranta@unipv.it

**Data Availability Statement:** All relevant data are within the paper and its Supporting Information files.

## Abstract

The aim of the Mont Blanc Study was to investigate the relationship between intraocular pressure (IOP), central corneal thickness (CCT), and altitude in healthy subjects. Thirty-three eyes of 33 healthy volunteers (mean age: 24.8 years, 17 females) had their IOP measured with Perkins and I-Care tonometers and their CCT using ultrasound pachymetry at three locations in Italy with different altitudes: Pavia, (PV), 77 meters above sea level (a.s.l); Courmayeur (CM), 1300 meters a.s.l; Pointe Helbronner (PH), 3466 meters a.s.l.). The measurements were performed at 9 am, 11 am, 1 pm and 3 pm (±30') in indoor settings (mean temperature of 19˚C) in PV and PH. At 9 am, CCT and IOP were measured outdoor (mean temperature of -1.4˚C) at PH. The mean values of the IOP curve decreased from PV to PH with the Perkins ($p = 0.02$) and I-Care tonometers ($p = 0.001$). Instead, CCT increased upon ascension from PV to PH ($p = 0.01$), and from CM to PH ($p = 0.002$). When exposed to sub-zero temperature, the IOP increased ($p<0.001$), while the CCT did not change ($p = 0.30$). The results suggest that IOP significantly decreased and CCT significantly increased upon ascension from the sea level to higher altitudes.

## Introduction

The increasing popularity of recreational activities such as mountain trekking and skiing has raised interest in the effects that hypoxic and hypobaric conditions have on the human body. In particular, intraocular pressure (IOP) changes are clinically important for glaucomatous patients and persons at higher risk of developing glaucoma, because IOP is a key risk factor for glaucoma onset and progression. [1]

Previous studies that investigated the relationship between IOP and altitude produced contrasting results. While several papers have shown that IOP increases with increasing altitude,

**Funding:** The authors received no specific funding for this work. The contribution of the author Dr. Alice C. Verticchio Vercellin and Dr. Ivano Riva was supported by Fondazione Roma and by the Italian Ministry of Health. The funders had no role in study design, data collection and analysis, decision to publish, or preparation of the manuscript.

**Competing interests:** The authors have declared that no competing interests exist.

[2, 3] others have suggested an inverse correlation between IOP and altitude. [4–7] These conflicting results may be due to several methodological issues that could have influenced the recordings at different altitudes, e.g. the physical effort performed by the subjects to reach higher altitudes, [2, 6, 7] the fact that central corneal thickness (CCT) measurements may have affect the measured IOP values, [3–5] and different temperature conditions in which IOP was measured. [5]

The aim of the 'Mont Blanc Study' was to evaluate the relationship between altitude and IOP, measured with two different tonometers, and CCT. The systemic parameters (arterial blood pressure, BP, heart rate, HR, arterial oxygen saturation, $SaO_2$) were also assessed.

## Materials and methods

The Mont Blanc Study was a prospective, observational cohort study. The Ethics Committee of the Fondazione IRCCS Policlinico San Matteo, Pavia, Italy, approved the procedure in accordance with the 1964 Declaration of Helsinki. Subjects were asked to fill out a general health questionnaire and to sign an informed consent form.

Thirty-three healthy volunteers (mean age: 24.8 ± 3.3, 17 female) were recruited. Exclusion criteria were: any type of systemic and/or ocular disease; positive family history for glaucoma; arterial BP>140/90 mmHg measured during the enrollment visit; any systemic medication that could influence IOP and BP values. The test eye was determined based on the inclusion and exclusion criteria. For each study participant, if both eyes qualified, one eye was randomly assigned as the observational study eye, and the same eye was utilized throughout the study. All subjects were asked to abstain from coffee and alcoholic beverages at least in the 12 hours before measurements. Smokers were asked not to smoke at least in the 30 minutes before measurements. [8, 9].

The subjects enrolled in the study did not perform any physical effort nor underwent any acclimatization process, and measurements of CCT and IOP were assessed and collected at the same time point and in the same environmental temperature.

In the present study IOP was measured in healthy volunteers at 77 m above sea level (a.s.l.) (Pavia, PV, Italy), at 1,300 m a.s.l. (Courmayeur, CM, Italy) and 3,466 m a.s.l. (Pointe Helbronner, PH, Mont Blanc Mountain, Italy). The ascent from CM was performed via a high-speed cableway, which took 15 minutes to reach PH. IOP and CCT, and the aforementioned systemic parameters, were measured in closed environment at constant temperature at the three different altitudes (IN conditions). IOP and CCT were also measured in the open air at PH (OUT conditions), in order to investigate the effect of cold on these parameters.

### Enrollment visit

The medical examination at the enrollment visit was performed in PV and consisted of a complete ophthalmic examination that included: determination of axial length (AL) and corneal curvature assessed by optical biometry (IOLMaster 500, Carl Zeiss Meditec AG, Germany); and evaluation of the optic nerve head (ONH) parameters and retinal nerve fiber layer (RNFL) and ganglion cell complex (GCC) thickness assessed by spectral domain optical coherence tomography (iVue SD-OCT system, Optovue, Inc., Fremont, CA). CCT was assessed by ultrasound pachymetry (PachMATE2) and IOP by both applanation tonometry (Perkins MK2, Clement and Clarke) and rebound tonometry (I-Care TAO1i tonometer, Icare Finland Oy), under topical anaesthesia with Oxybuprocaine drops. A tonometric curve has been carried out with four IOP measurements at 9 am (± 30 min), 11 am (± 30 min), 1 pm (± 30 min) and 3 pm (± 30 min). At 9 am BP, HR and $SaO_2$ were also recorded. All parameters were measured with the subject seated for at least 5 minutes.

## SkyWay data collection

Measurements of CCT, IOP, BP, HR and $SaO_2$ of the eligible subjects were assessed at 9 am (± 30 min) in the indoor environment of the CM Station (mean temperature: 19˚C). These measurements, along with the IOP curve, were repeated at PH, which was reached by the 'Skyway Mont Blanc cable car' in 15 minutes. In addition, environmental temperature, atmospheric pressure, and humidity were also measured at PH. CCT and IOP were also assessed in the open air in a sub-zero temperature (-1.4˚C) at PH at 9.00 am (± 30 min).

## Statistical analysis

Quantitative variables were normally distributed (Shapiro test) and were described as Mean and Standard Deviation (SD); qualitative variables were summarized using count and percentages, while differences were analysed using the chi square test. For quantitative variables, comparisons between two groups were carried out with the t-test for independent observations, while analysis of variance (ANOVA) with Bonferroni-adjusted p values was used to compare continuous variables among more than two groups.

To take in account different factors that could affect the association between IOP and altitude, multilevel mixed models were fitted using IOP as dependent variables, place (PV or PH) and time as independent variables in the fixed part of the models, and patients as random factors. Models were then adjusted for CCT. Adjusted differences and 95% confidence intervals (95%CI) were computed. Interaction of altitude and time of the day to assess whether the time of the day influenced the difference was tested and excluded.

Post hoc comparisons were also performed at each time point (Bonferroni correction). $P < 0.05$ was considered statistically significant. All tests were two-sided. Data analysis was performed using the STATA statistical package (release 14.1, 2015, Stata Corporation, College Station, Texas, USA).

## Results

One eye from each of 33 healthy volunteers (mean age: 24.8 ± 3.3 years, 17 females) was considered in the analysis. The measured parameters (RNFL and GCC thickness, ONH parameters) were within normal limits for all subjects (detailed results not shown). The mean AL was 23.58 mm [SD: 0.68]. Tonometric measures are summarized in Table 1.

### Effect of altitude on IOP

Altitude was significantly associated with differences in the IOP measures (Perkins $p = 0.19$, I-Care $p < 0.001$), while accounting for the time of the day and CCT in a mixed regression model.

As shown in Table 1, the mean IOP values decreased significantly and similarly from PV to PH with both Perkins ($p = 0.020$) and I-Care ($p = 0.001$) tonometers. Also, IOP was shown to decrease from PV to CM with both instruments, though only measures performed with the I-Care instrument retained statistical significance ($p = 0.002$); finally, no significant difference was observed for changes from CM to HB with any of the two instruments.

Tonometric values decreased over the time of the day both with the Perkins ($p = 0.002$) and the I-Care ($p = 0.016$) instruments (Table 1).

However, the time of the day did not modify the size of the difference between altitudes tonometric measures (Perkins, $p$ for interaction = 0.39 and I-Care, $p$ for interaction = 0.91).

**Table 1. Comparison of Intraocular Pressure (IOP) (mean values ± standard deviation, SD) measured in mmHg at four time point (9 am (± 30 min), 11 am (± 30 min), 1 pm (± 30 min) and 3 pm (± 30 min)).**

| Hour | Pavia[1] mean (SD) | Courmayeur[2] mean (SD) | Pointe Helbronner[3] mean (SD) | Adjusted mean difference (95%CI)* | p-value (Bonferroni) |
|---|---|---|---|---|---|
| **Perkins** | | | | | |
| 9 | 11.94(±2.47) | 11.36(±3.53) | 11.58(±2.72) | [2vs1]−0.80 (-2.10 to 0.50) | 0.418 |
| 11 | 11.03(±2.36) | | 10.79(±2.63) | 3vs1−0.82 (-1.54 to -0.10) | 0.020 |
| 13 | 11.27(±2.67) | | 10.00(±3.21) | 3vs2−0.02 (-1.32 to 1.29) | 1.000 |
| 15 | 10.88(±2.47) | | 9.48(±2.86) | | |
| **I-Care** | | | | | |
| 9 | 15.27(±3.38) | 13.80(±3.10) | 14.61(±2.94) | [2vs1]−1.58 (-2.69 to -0.48) | 0.002 |
| 11 | 14.44(±3.42) | | 13.53(±3.39) | 3vs1−0.89 (-1.51 to -0.28) | 0.001 |
| 13 | 14.39(±3.39) | | 13.56(±3.46) | [3vs2] 0.69 (-0.41 to 1.79) | 0.403 |
| 15 | 14.61(±4.01) | | 13.44(±2.91) | | |

The IOP recordings were assessed by Perkins and I-Care tonometers. The measurements were obtained at different altitudes: at 77 m a.s.l. (Pavia, 1), at 1,300 m a.s.l. (Courmayeur, 2) and at 3,466 m a.s.l. (Pointe Helbronner, 3).

*adjusted for time of the day & central corneal thickness.

## Central corneal thickness upon ascension

CCT increased upon ascension from PV to PH (PV vs PH, Mean [SD]: 559.45 μm [25.35] vs 563.22 μm [26.05]; $p$ = 0.01, Fig 1). An increase in CCT was also found between CM and PH (CM vs PH, 558.86 μm [24.20] vs 563.22 μm [6.05]; $p$ = 0.002). No significant difference was found between PV and CM (PV vs CM, 559.45 μm [25.35] vs 558.86 μm [24.20]; $p$ = 0.62).

## Effect of a sub-zero environmental temperature on IOP and CCT

When exposed to a sub-zero temperature, the IOP increased (IN vs OUT, Mean [SD]: 14.29 mmHg [2.2] vs 16.98 mmHg [3.71]; $p$<0.001, Fig 2A), while CCT was similar in the two conditions (IN vs OUT, 563.22 μm [26.05] vs 565.20 μm [27.21]; $p$ = 0.30, Fig 2B).

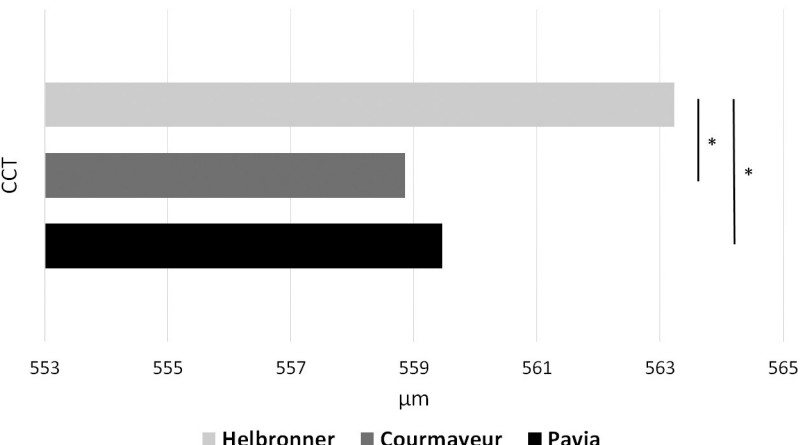

**Fig 1. Central corneal thickness at three different altitudes.** Changes of central corneal thickness measured in μm upon ascension from Pavia (PV, black bar), to Courmayeur (CM, dark grey bar), and Pointe Helbronner (PH, light grey bar). CCT increased in PH with respect to both PV and CM. *P-value indicates a statistically significant difference (p<0.05).

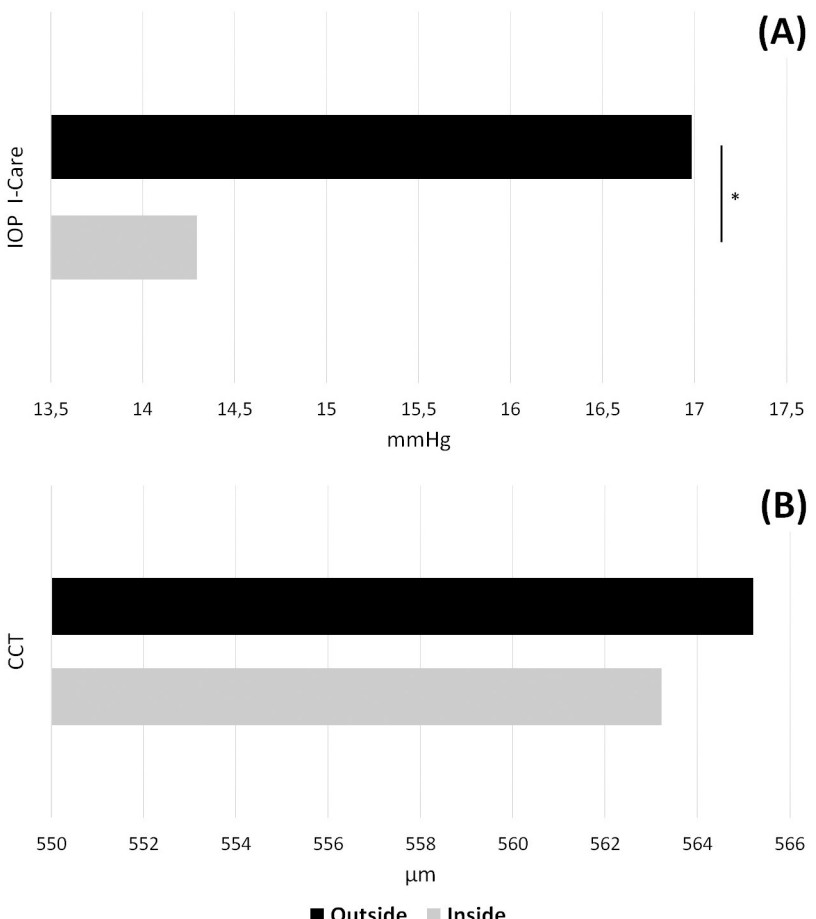

**Fig 2. Effect of sub-zero temperature on intra ocular pressure and central corneal thickness.** Comparison of intraocular pressure and central corneal thickness at Pointe Helbronner (3,466 m above sea level) measured in a closed environment (19°C) and in the open air (-1°C). Outside values are represented with black bars and inside values are represented with light grey bars. Statistics found no difference in CCT, but a significant effect of cold on IOP. *P-value indicates a statistically significant difference (p<0.05).

## Correlation between IOP and systemic parameters

Clinical parameters of diastolic BP (DBP), systolic BP (SBP) and HR significantly increased, and arterial oxygen saturation (SaO$_2$) significantly decreased at high altitudes compared to sea level. An overview of all clinical data is shown in Table 2.

**Table 2. Mean diastolic and systolic blood pressure, heart rate and arterial oxygen saturation at three altitudes.**

|  | DBP (mmHg) | SBP (mmHg) | HR (beats/min) | SaO$_2$(%) |
|---|---|---|---|---|
| **PH** | 77.8±7.6 | 122.7±7.6 | 79.3±9.9 | 88.2±2 |
| **CM** | 79±7.1 | 126.9±8.9 | 75.7±11.6 | 96.8±1.4 |
| **PV** | 67.5±5.5 | 115.9±7.4 | 68.6±8.4 | 99 |

Mean values (±standard deviation, SD) of systemic parameters measured at three different altitudes: PH (3,466 m a.s. l., Pointe Helbronner), CM (1,300 m a.s.l., Courmayeur) and PV (77 m a.s.l. Pavia). DBP, diastolic blood pressure; HR, heart rate; SaO$_2$, arterial oxygen saturation; SBP, systolic blood pressure.

A significant IOP increase of 0.8 mmHg (Perkins) or 0.6 mmHg (I-Care) along a 10 mmHg increase of DBP was observed from PV to PH ($p<0.001$ and $p = 0.018$ respectively). Moreover, a direct relationship was found between IOP and $SaO_2$, with an increase of 0.12 mmHg (Perkins) or 0.30 mmHg (I-Care) for each unit of percentage of $SaO_2$ ($p = 0.050$ and $p = 0.006$ respectively). Present results also showed an IOP reduction of -0.3 mmHg (Perkins) and -0.5 mmHg (I-Care) for 10 bpm increase in HR (p = 0.050 and p = 0.038 respectively).

## Discussion

The Mont Blanc Study was conducted in order to evaluate the relationship between IOP, CCT and systemic circulatory parameters (SBP, DBP, $SaO_2$) in a group of healthy subjects at different altitudes. The present data showed an inverse correlation between IOP and altitude: the mean values of the tonometric curve decreased at PH compared to PV. The decrease in IOP with high altitude corresponded to a decrease in DBP and $SaO_2$. The reduction in IOP was mirrored by an increase of CCT upon ascension from PV to PH.

### Effect of altitude on IOP and CCT

The diurnal curves recorded at different altitudes (PV and PH, Table 1) showed the highest IOP values at 9 am both with Perkins and I-Care, in accordance with previous literature. [10–12] Upon ascension from the sea level to high altitudes, the mean values of the tonometric curve significantly decreased. Present results were obtained overcoming the limitations of the previous studies investigating the relationship between IOP, CCT and altitude. In details, the studied subjects did not perform any physical effort, nor underwent any acclimatization process, and IOP and CCT measurements were assessed at the same time of the day at different altitudes and with the same environmental temperature. The magnitude of IOP changes at high altitude found in the present paper is in accordance with previous literature. [2, 3, 13, 14] The approximately one mmHg of IOP decreased at high altitude could be considered relatively small. However, taking into account that each millimetre of mercury of IOP elevation might increase by about 20% the risk of developing glaucoma, [1, 13] while each mmHg of IOP reduction may decrease the risk of glaucoma progression by about 10%, [15] the IOP changes induced by altitude can have clinical significance. Concerning the physiological mechanisms causing the IOP reduction at PH, it could be hypothesized that a relationship between IOP and hypoxic conditions exists; depleted oxygen supply to the non-pigmented ciliary epithelium and hence to decreased aqueous humor production induce a decline in IOP at high altitudes, where systemic hypoxia occurs. [4] Indeed, part of the aqueous humor production depends on a unidirectional transport system of solutes, passively followed by water through osmosis. [16] In details, the carbonic anhydrase catalyzes the rapid conversion of carbon dioxide and water to bicarbonate and protons ($CO_2 + H_2O -> H_2CO_3 -> H + HCO_3$). Hydrogen ion is exchanged for sodium, which is expelled through sodium/potassium phosphatase. The high concentration of sodium and bicarbonate in the intercellular spaces leads to the shift in the posterior chamber of water by osmosis. The human body reacts to hypoxia with adaptive measures, among which the increase in respiratory depth. The hyperventilation decreases the carbonic acid ($H_2CO_3$) production, by producing a decrease in $CO_2$. As a consequence, H + concentration is reduced in the posterior chamber, leading to a decrease of the osmotic gradient between capillaries in the ciliary body and the posterior chamber. Therefore, we can speculate that this mechanism induced the decrease in aqueous humor production and of IOP at high altitudes. [17] It is important to highlight that both the Perkins and I-care tonometers showed a decrease of IOP values at high altitude in the study subjects. However, the I-Care

tonometer is known to overestimates IOP compared to the Perkins, [18] thus the difference between the IOP measurements assessed by the two instruments at each time point.

An increase of CCT was recorded at PH compared to PV, in accordance with previous literature. [2–4, 7] The hypoxic conditions found at high altitude might hinder corneal endothelial function, thus inducing corneal edema and therefore an increase of CCT. [2]

### Effect of low environmental temperature on IOP and CCT

With cold, IOP increased (p<0.001) while CCT did not undergo any significant change. Ortiz et al. [19] evaluated the effect of a stream of cold air (-19˚C) directed toward a closed eye. After 40 minutes of exposure to continuous stream of cold air a decrease of IOP was found. The different results obtained in the present paper may be explained by the fact that the effects of low environmental temperature at PH influence the whole body, and was not directed only toward the eye. Indeed, cold is known to induce a sympathetic drive, [20] with an increase of norepinephrine release that rises the IOP. [21]

### Correlation between IOP and systemic parameters

The recorded levels of SaO$_2$ were lower at PH when compared to both PV and CM, in accordance with literature. [22] The decrease in SaO$_2$ is explained by the fall of partial pressure of oxygen in the breathing air at high altitudes. Although the percentage of oxygen in inhaled air is constant at different altitudes, the fall in atmospheric pressure at higher altitudes decreases the partial pressure of inhaled oxygen and hence the driving pressure for gas exchange in the lungs. SBP, DBP and HR increased in CM and at PH compared to PV. These findings are in line with the existing literature. [20, 23–26] The rise in BP upon ascension is due to the effect of oxygen deprivation in increasing the activity of the sympathetic nervous system. [24] HR increases as an adaptive measure to the fall in partial pressure of oxygen at high altitude in order to keep up with oxygen delivery to the tissues. [27]

A direct relationship was found when comparing DBP and IOP, in accordance with previous papers. [28, 29] Moreover, the present study also showed a direct relationship between IOP and SaO$_2$, in accordance with Pavlidis et al., 2006 [6] and Bosch et al., 2010 [4]. On the contrary, Nazari et al., 2013 [5] did not find any correlation between these two parameters. These differences can be explained by methodological differences as well as dissimilarities in environmental conditions in which these studies were performed.

### Limitations of the study

The Mont Blanc study has certain limitations. The first one is that the tonometers used in the study have not been validated for use at high altitudes. Nonetheless, other studies carried out at comparable altitudes have utilized the same instruments. Further studies to validate the use of such instruments at high altitude are indeed required. A second limitation is the lack of data on the acclimatization process: it would be important to assess the influence of acclimatization on IOP and CCT measurements in healthy volunteers. Thirdly, the Mont Blanc study volunteers were healthy subjects with a mean age of 24.8 years. Glaucomatous subjects are older and the pathophysiological mechanism underlying their condition may affect IOP and CCT at high altitude differently compared to healthy subjects.

## Conclusions

The Mont Blanc study showed that IOP decreases and CCT increases upon ascension at high altitudes The Mont Blanc study findings suggest that high altitude, decreasing the IOP, might

represent a protective factor for the glaucomatous patients. In order to test this hypothesis and its clinical relevance, further studies are needed in order to investigate the influence of altitude in a glaucomatous population and the effect of acclimatization on IOP and CCT.

## Supporting information

**S1 Dataset.**
(XLS)

## Acknowledgments

The authors would like to thank the Skyway Monte Bianco (Funivie Monte Bianco S.p.a, Aosta Valley, Italy), the Regione Valle d'Aosta, and all the volunteers who participated in the study. The authors would like to thanks for their contribution Drs. Beatrice Montanaro, Massimiliano Manera and Giovanni Milano.

## Author Contributions

**Conceptualization:** Carlo Bruttini.

**Data curation:** Carlo Bruttini, Alice Verticchio Vercellin, Catherine Klersy.

**Formal analysis:** Carlo Bruttini, Catherine Klersy, Annalisa De Silvestri, Carmine Tinelli, Ivano Riva.

**Investigation:** Carlo Bruttini, Alice Verticchio Vercellin.

**Methodology:** Carlo Bruttini, Alice Verticchio Vercellin, Carmine Tinelli.

**Supervision:** Luciano Quaranta.

**Writing – original draft:** Carlo Bruttini, Alice Verticchio Vercellin.

**Writing – review & editing:** Carlo Bruttini, Alice Verticchio Vercellin, Catherine Klersy, Ivano Riva, Francesco Oddone, Andreas Katsanos, Luciano Quaranta.

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
