## [Decision Letter · Decision Letter 0]

4 May 2020

PONE-D-20-07934

The Mont Blanc Study: the effect of Altitude on Intra Ocular Pressure and Central Corneal Thickness

PLOS ONE

Dear Dr. Quaranta,

Thank you for submitting your manuscript to PLOS ONE. After careful consideration, we feel that it has merit but does not fully meet PLOS ONE’s publication criteria as it currently stands. Therefore, we invite you to submit a revised version of the manuscript that addresses the points raised during the review process.

Both learned reviewers have offered criticisms of the manuscript. However, a reviewer have pointed out potential inclusion of data and several questions that need to be carefully addressed by making appropriate revisions. 

We would appreciate receiving your revised manuscript by Jun 18 2020 11:59PM. To enhance the reproducibility of your results, we recommend that if applicable you deposit your laboratory protocols in protocols.io, where a protocol can be assigned its own identifier (DOI) such that it can be cited independently in the future. For instructions see: http://journals.plos.org/plosone/s/submission-guidelines#loc-laboratory-protocols

We look forward to receiving your revised manuscript.

Kind regards,

Sanjoy Bhattacharya

Academic Editor

PLOS ONE

Reviewers' comments:

Reviewer's Responses to Questions

**Comments to the Author**

1. Is the manuscript technically sound, and do the data support the conclusions?

Reviewer #1: Yes

Reviewer #2: Partly

2. Has the statistical analysis been performed appropriately and rigorously? 

Reviewer #1: Yes

Reviewer #2: I Don't Know

3. Have the authors made all data underlying the findings in their manuscript fully available?

Reviewer #1: Yes

Reviewer #2: Yes

4. Is the manuscript presented in an intelligible fashion and written in standard English?

Reviewer #1: Yes

Reviewer #2: Yes

5. Review Comments to the Author

Reviewer #1: As you suggest acclimatization may play a role. The length of time subjects were in each environment prior to testing should be noted. How was the test eye determined ? You need to state that the same eye was utilized throughout the study. A comparison of the length of time at each environment with real life time of exposures and potential effects should be included.

Reviewer #2: This is an interesting study. Published reports comparing IOP and altitude have shown contrasting results. The authors of this study found an inverse correlation between altitude and IOP, while CCT correlated with ascension from sea level. There are a few issues that I have with the data as presented.

The following are some comments and questions:

1.) In figure one there I cannot see a line for Courmayeur. There is only a triangle shown at the 9:00 am mark.

2.) Table 2 header says I-Care tonometer was used, while the figure legend says Perkins was used.

3.) Where are the mean values coming from detailed in the paragraph starting at line 125? Is this mean across all time points? If so, that is not seeming to match up with the table. Please add a separate column in the table detailing the means from this specific analysis that was discussed in the aforementioned paragraph. Also, was the mean value used in that calculation from all time points? If so, did authors measure the value of these 33 volunteers 4 separate times, then average that? Or did they average each individual across all time points?

4.) The authors mention in the introduction that CCT may have an effect on IOP values (line 33-34). However, in the work they did not explore this further. The authors should examine the IOP across different attitudes while controlling for CCT values. They already have this data so it should not be difficult.

5.) One central issue that needs to be discussed is why the large discrepancy between the tonometers between PH and PV in Tables 1 and 2. We see significant differences with I-Care at three time points, yet only one significant difference with Perkins (3pm). Which table are we to believe? We see only a common significant change at 3 pm. This should be discussed further.

6.) Following from point 5. There seems to be significant changes across time points. Since authors are presenting this data, that needs to be discussed further. What influence does time have on IOP measurements at these varying elevations.

6. PLOS authors have the option to publish the peer review history of their article (what does this mean?). If published, this will include your full peer review and any attached files.

Reviewer #1: No

Reviewer #2: No

---

## [Author Response · Author response to Decision Letter 0]

24 Jun 2020

We would like to thank the editor and reviewers of PLOS One for your indications. Based on the comments we have edited the manuscript as outlined below:

Review Comments to the Author:

Reviewer #1: As you suggest acclimatization may play a role. The length of time subjects were in each environment prior to testing should be noted. How was the test eye determined ? You need to state that the same eye was utilized throughout the study. A comparison of the length of time at each environment with real life time of exposures and potential effects should be included.

We thank the reviewer for the comments. The test eye was determined based on the inclusion and exclusion criteria. For each study participant, if both eyes qualified, one eye was randomly assigned as the observational study eye, and the same eye was utilized throughout the study. The measurements were taken on two different days, one day in Pavia and one day on the Mont Blanc. The study measurements were taken in Pavia at four different time points (at 9 am (± 30 min), 11 am (± 30 min), 1 pm (± 30 min) and 3 pm (± 30 min)), and the subjects spent the time from 9 am (± 30 min) to 3 pm (± 30 min) in the indoor hospital settings of the University Eye Clinic of Pavia. The second day of the study experiment took place at the Skyway Mount Blanc. In details, the subjects underwent the measurement at 9 am (± 30 min) in the indoor settings of the Courmayeur Station (mean temperature: 19°C). Immediately after, the subject took the ‘Skyway Mont Blanc cable car’ and they reached in 15 minutes the Pointe Helbronner Station, where the measurements were repeated at the same time points evaluated in Pavia (at 9 am (± 30 min), 11 am (± 30 min), 1 pm (± 30 min) and 3 pm (± 30 min)). When the subjects reached the Pointe Helbronner Station, the measurement of the first time point (9 am (± 30 min)) was taken both indoor and in the open air in a sub-zero temperature; immediately after the outdoor measurement all the subjects went back indoor, and they spent the time from 9 am (± 30 min) to 3 pm (± 30 min) in the indoor settings of the Pointe Helbronner Station. Since at least 1 to 3 days are required for the body to undergo the physiological changes characteristics of the acclimatization process, further studies are needed to evaluate this aspect, that was beyond the purpose of our study, as highlighted in the “limitations of the study” section of the manuscript. We edited the revised manuscript to clarify the methods per your comments, thank you. 

Reviewer #2: This is an interesting study. Published reports comparing IOP and altitude have shown contrasting results. The authors of this study found an inverse correlation between altitude and IOP, while CCT correlated with ascension from sea level. There are a few issues that I have with the data as presented.

The following are some comments and questions:

1.) In figure one there I cannot see a line for Courmayeur. There is only a triangle shown at the 9:00 am mark.

We thank the reviewer for the indication. The Reviewer is right. There is only one measurement in Courmayeur at 9 am; Figure 1 has been removed in the revised manuscript, these data are now detailed in Table 1. 

2.) Table 2 header says I-Care tonometer was used, while the figure legend says Perkins was used.

Thank you for the comment. In the revised manuscript, Table 2 has been removed. We added a comparison of IOP values assessed by both Perkins and I-Care tonometers in Table 1.

3.) Where are the mean values coming from detailed in the paragraph starting at line 125? Is this mean across all time points? If so, that is not seeming to match up with the table. Please add a separate column in the table detailing the means from this specific analysis that was discussed in the aforementioned paragraph. Also, was the mean value used in that calculation from all time points? If so, did authors measure the value of these 33 volunteers 4 separate times, then average that? Or did they average each individual across all time points?

We thank the Reviewer for this comment. We amended the analysis in the revised manuscript in order to address the Reviewer concerns and we have adjusted for CCT. We added these data in Table 1 of the revised manuscript.

4.) The authors mention in the introduction that CCT may have an effect on IOP values (line 33-34). However, in the work they did not explore this further. The authors should examine the IOP across different attitudes while controlling for CCT values. They already have this data so it should not be difficult.

The multilevel mixed models was adjusted for CCT. This new analysis has been introduced in the revised MS per your comment, thank you. 

5.) One central issue that needs to be discussed is why the large discrepancy between the tonometers between PH and PV in Tables 1 and 2. We see significant differences with I-Care at three time points, yet only one significant difference with Perkins (3pm). Which table are we to believe? We see only a common significant change at 3 pm. This should be discussed further.

We thank the reviewer for the comment. In the revised manuscript, a new statistical analysis was performed, and we included the new data in the “results” section and Table 1 of the revised manuscript. As shown in Table 1 of the revised manuscript, the mean IOP values decreased significantly and similarly from Pavia to Pointe Helbronner with both the Perkins (p=0.020) and the I-Care (p=0.001) tonometers. Also, IOP was shown to decrease from Pavia to Courmayeur with both instruments, though only measures performed with the I-Care instrument retained statistical significance (p=0.002); finally, no significant difference was observed for changes from Courmayeur to Pointe Helbronner with any of the two instruments. Tonometric values decreased over the time of the day both with the Perkins (p=0.002) and the I-Care (p=0.016) instruments (Table 1). Perkins tonometer is considered the ‘gold standard’ in handheld applanation tonometers, as the IOP measurements have been shown to be comparable to those of Goldmann applanation tonometer. On the other hand, the IOP measurements assessed by I-Care have shown a good repeatability, and this type of tonometer is used widely for its several advantages, such as the fact that it does not require the use of anesthetic eye drops and fluorescein dye, and it measures the IOP with reduced discomfort for the subjects. It is important to highlight that I-Care tonometer is known to overestimates IOP compared to the Perkins (Ting SL, Lim LT, Ooi CY, Rahman MM. Comparison of Icare Rebound Tonometer and Perkins Applanation Tonometer in Community Eye Screening. Asia Pac J Ophthalmol (Phila). 2019;8(3):229‐232. doi:10.22608/APO.2018433), thus the difference between the IOP measurements assessed by the two instruments at each time point. Importantly, both the Perkins and I-Care tonometers showed a decrease of IOP values at high altitude in the study subjects, thus suggesting the potential use of both of them in settings where the need of a handheld tonometer is required. The revised manuscript has been edited per your comments, thank you. 

6.) Following from point 5. There seems to be significant changes across time points. Since authors are presenting this data, that needs to be discussed further. What influence does time have on IOP measurements at these varying elevations.

Both the differences between altitudes and the differences between time of the day were explored while adjusting for CCT. This is now better detailed in Table 1 and in the text of the revised manuscript. Thank you for the comment.

---

## [Decision Letter · Decision Letter 1]

24 Jul 2020

The Mont Blanc Study: the effect of Altitude on Intra Ocular Pressure and Central Corneal Thickness

PONE-D-20-07934R1

Dear Dr. Quaranta,

We’re pleased to inform you that your manuscript has been judged scientifically suitable for publication and will be formally accepted for publication once it meets all outstanding technical requirements.

Kind regards,

Sanjoy Bhattacharya

Academic Editor

PLOS ONE

Additional Editor Comments (optional):

Reviewers' comments:

Reviewer's Responses to Questions

**Comments to the Author**

1. If the authors have adequately addressed your comments raised in a previous round of review and you feel that this manuscript is now acceptable for publication, you may indicate that here to bypass the “Comments to the Author” section, enter your conflict of interest statement in the “Confidential to Editor” section, and submit your "Accept" recommendation.

Reviewer #1: All comments have been addressed

Reviewer #2: All comments have been addressed

2. Is the manuscript technically sound, and do the data support the conclusions?

Reviewer #1: Yes

Reviewer #2: Yes

3. Has the statistical analysis been performed appropriately and rigorously? 

Reviewer #1: Yes

Reviewer #2: Yes

4. Have the authors made all data underlying the findings in their manuscript fully available?

Reviewer #1: Yes

Reviewer #2: Yes

5. Is the manuscript presented in an intelligible fashion and written in standard English?

Reviewer #1: Yes

Reviewer #2: Yes

6. Review Comments to the Author

Reviewer #1: (No Response)

Reviewer #2: Authors have adequately addressed concerns of this reviewer. I believe that this article should be published by PLOS one.

7. PLOS authors have the option to publish the peer review history of their article (what does this mean?). If published, this will include your full peer review and any attached files.

Reviewer #1: No

Reviewer #2: No

---

## [Editor Report · Acceptance letter]

29 Jul 2020

PONE-D-20-07934R1 

The Mont Blanc Study: the effect of Altitude on Intra Ocular Pressure and Central Corneal Thickness 

Dear Dr. Quaranta:

I'm pleased to inform you that your manuscript has been deemed suitable for publication in PLOS ONE. Congratulations! Your manuscript is now with our production department. 

Kind regards, 

on behalf of

Dr. Sanjoy Bhattacharya 

Academic Editor

PLOS ONE